# Green Composites from Partially Bio-Based Poly(butylene succinate-*co*-adipate)-PBSA and Short Hemp Fibers with Itaconic Acid-Derived Compatibilizers and Plasticizers

**DOI:** 10.3390/polym14101968

**Published:** 2022-05-12

**Authors:** Celia Dolza, Eloi Gonga, Eduardo Fages, Ramon Tejada-Oliveros, Rafael Balart, Luis Quiles-Carrillo

**Affiliations:** 1Textile Industry Research Association (AITEX), Plaza Emilio Sala 1, 03081 Alcoy, Spain; cdolza@aitex.es (C.D.); egonga@aitex.es (E.G.); efages@aitex.es (E.F.); 2Institute of Materials Technology (ITM), Universitat Politècnica de València (UPV), Plaza Ferrándiz y Carbonell 1, 03801 Alcoy, Spain; rbalart@mcm.upv.es (R.B.); luiquic1@epsa.upv.es (L.Q.-C.)

**Keywords:** BioPBSA, hemp natural fibers, green composites, itaconic acid, agricultural waste valorization

## Abstract

In this work, green composites have been developed and characterized using a bio-based polymeric matrix such as BioPBSA and the introduction of 30 wt.% short hemp fibers as a natural reinforcement to obtain materials with maximum environmental efficiency. In order to increase the interfacial adhesion between the matrix and the fiber to obtain better properties in the composites, a reactive extrusion process has been carried out. On the one hand, different additives derived from bio-based itaconic acid have been added to the BioPBSA/HEMP composite, such as dibutyl itaconate (DBI) and a copolymer of PBSA grafted with itaconic acid (PBSA-*g*-IA). On the other hand, a different copolymer of PBSA grafted with maleic anhydride (PBSA-*g*-MA) was also tested. The resulting composites have been processed by injection-molding to obtain different samples which were evaluated in terms of mechanical, thermal, chemical, dynamic-mechanical, morphological and wettability and color properties. In relation to the mechanical properties, the incorporation of hemp fibers resulted in an increase in the stiffness of the base polymer. The tensile modulus of pure BioPBSA increased from 281 MPa to 3482 MPa with 30% fiber. The addition of DBI shows a remarkable improvement in the ductility of the composites, while copolymers with IA and MA, generate mechanically balanced composites. In terms of thermal properties, the incorporation of hemp fiber and compatibilizing agents led to a reduction in thermal stability. However, from the point of view of thermomechanical properties, a clear increase in rigidity is achieved throughout the temperature range studied. As far as the color of the samples is concerned, the incorporation of hemp generates a typical color, while the incorporation of the compatibilizing agents does not modify this color excessively. Finally, the introduction of lignocellulosic fibers greatly affects water absorption and contact angle, although the use of additives helped to mitigate this effect.

## 1. Introduction

Plastics are a basic element of our daily life. They are widely used due to their versatility, durability, ease of processing and low cost [1]. Plastics are polymers that are chemically synthesized from fossil resources such as petroleum and which contain long monomer chains. As a consequence of its widespread use, there are large amounts of waste that must be properly managed to prevent these wastes from ending up in landfills or spread throughout the natural environment [2]. As a result of the intensive use of non-compostable or non-recyclable petroleum-derived materials, there is great concern about environmental pollution, greenhouse gas emissions, human health, and the depletion of fossil resources [3,4]. In addition, plastics are the main cause of pollution and toxicity of the aquatic ecosystem of rivers, seas, and oceans [5]. These are the main factors that have promoted the study and development of environmentally friendly polymeric materials with good properties in order to replace traditional petroleum-based plastics [6,7]. The development of biorefinery has been a major step forward in the production of the corresponding bio-based monomers or building blocks for the production of polyester, polyurethanes and polyamides [8,9,10], among others. As a result, the production of biobased polymers, as an alternative to fossil-sources, is currently in a growing state [11]. Bio-polyesters such as poly(butylene succinate) (PBS), poly[(butylene adipate)-*co*-terephthalate] (PBAT), poly(3-hydroxybutyrate-*co*-3-hydroxyvalerate) (PHBV), and poly[(butylene succinate)-*co*-adipate] (PBSA) have high ductility, good processability and the capacity to biodegrade [12]. In this context, it is currently possible to obtain PBSA from renewable natural sources, thus obtaining a material with high environmental efficiency.

Bio-based PBSA is a random copolymer of poly(butylene succinate) (PBS); it is synthesized by polycondensation of 1,4-butanediol in the presence of succinic and adipic acids with relatively low production cost, satisfactory mechanical properties and also has excellent processability [13], being able to be conformed by extrusion, injection, blow molding, in addition to obtaining filaments, multifilaments and films [14,15,16,17]. Due to these characteristics, PBSA is a thermoplastic polyester with great potential. However, it has limited thermal stability and gas barrier properties, which limit it in certain applications. [18]. To solve, or at least mitigate, this problem, several improvement strategies have been proposed in recent years, such as binary or ternary blending with other polymers, the development and introduction of copolymers, and the introduction of reinforcements or fillers, for example [19]. Currently, the so-called circular economy is growing, and its objective is to establish a loop in which waste could be reused to obtain new products, in order to reduce the requirement of raw materials and energy necessary to obtain them. In this context, the revalorization of by-products from industries or agroforestry is being applied to obtain natural fiber reinforced plastics (NFRP) [20,21] and wood–plastic composites (WPC) [22,23]. Natural fibers could play an important role in the development of biodegradable composites with improved properties and minimize existing environmental problems. Research has shown that the incorporation of fibers from husks, stems, leaves and forest residues into polymeric matrices has improved their mechanical properties [24,25,26].

One of the most interesting natural fibers is hemp fiber (*Cannabis sativa* L.), since it is an agricultural residue that is widely available, economical, and capable of reducing the cost of the polymer in which it is introduced. Although it is necessary to improve the affinity between the hemp fibers and the polymer matrix, there are several procedures such as the treatment of the fibers [27], or the use of additives (plasticizers and compatibilizers) [28], or the introduction of nanoparticles [29]. As a result of these strategies it is possible to produce components with a wide range of applications, such as indoor and outdoor furniture, manufacturing of automotive parts [30], building structures [31,32], or textile applications [33]. In particular, hemp fiber has been used effectively as a reinforcement in polymeric matrices such as polyethylene (PE), polypropylene (PP) and even polylactic acid (PLA) [34,35,36].

The use of compatibilizers is particularly interesting, because the complex treatments to be applied to fillers are not required. Due to their versatility and ease of use, a wide variety of these additives are now available, and new strategies are being developed to improve their synthesis and efficiency. In this respect, itaconic acid (IA) is attracting the interest of researchers and industries due to its natural origin, and it is the most easily produced bio-based unsaturated monomer by fermentation [37]. Itaconic acid (methylene succinic acid) is a crystalline unsaturated dicarbonic acid in which a carboxyl group is conjugated to the methylene group. It can be produced cost-effectively from sustainable substrates and has the potential to replace petrochemicals in the future [38]. It is used as a base chemical for the production of various value-added products, such as monomers, copolymers and terpolymers with properties similar to those of acrylates. In this regard, dibutyl itaconate (DBI) is a biobased monomer derived from the esterification of IA and 1-butanol [39]. In addition, DBI is mainly utilized as a comonomer in the synthesis of bio-based elastomers [40].

The main objective of this work is the development of green composites from the incorporation of short hemp fibers from agroforestry waste into a PBSA bio-based polymeric matrix. The aim is to obtain maximum environmental efficiency and improved mechanical properties by introducing different compatibilizing additives. In order to improve the matrix–fiber interaction, different compatibilization strategies have been developed. The use of additives based on itaconic acid, such as DBI (dibutyl itaconate) and the development of copolymers grafted with maleic anhydride (MA) and itaconic acid (IA), has been another objective to follow during the development of the work due to the great novelty in this field. Finally, the effectiveness of these strategies has been evaluated by chemical, mechanical, morphological, thermal, thermomechanical and FTIR characterization, as well as visual and wetting characteristics.

## 2. Materials and Methods

### 2.1. Materials

The matrix polymer, BioPBSA with CAS number 152049-37-1, was supplied by PTTMCC Biochem Company (Chauchak, Bangkok 10900, Thailand). The manufacturer supplies this bio-based PBSA in white granules with a relative density of 1.20–1.30. The hemp fiber used as reinforcement was supplied by SCHWARZWÄLDER TEXTIL-WERKE (Schenkenzell, Germany). This fiber had an irregular cross section with an average size between 15–50 μm and a specific weight of 1.48–1.50 g/cm^3^.

Itaconic acid (IA) and maleic anhydride (MA) were used to make the grafted copolymers. A dicumyl peroxide (DCP) initiator was used to improve the reaction. The above-mentioned compounds were supplied by Sigma-Aldrich S.A. (Madrid, Spain). The most important characteristics of each of these compounds are described below. Itaconic acid (IA), CAS number 97-65-4, has a molecular weight of 130.10 g/mol, relative density of 1.573 g/cm^3^ at 25 °C and water solubility of 77.49 g/L at 20 °C. Maleic anhydride (MA), CAS number 33225-51-3, has a specific gravity of 100.07 g/mol and a purity of 98%. Both AI and MA were supplied in white powder form. Dicumyl peroxide (DCP), CAS number 80-43-3, has a specific gravity of 270.37 g/mol and a purity of 98%, with an appearance of crystals or flakes with a white to pale yellow color.

### 2.2. Sample Preparation

Due to the moisture absorption capacity of the materials used, specifically BioPBSA and hemp fibers, they were previously dried for 48 h in a dehumidifying dryer MDEO (Industrial Marsé, Barcelona, Spain), in order to avoid hydrolysis in the processing of the mixtures. The compositions were mixed and homogenized and then introduced into a co-rotating twin screw extruder, Construcciones Mecánicas Dupra, S.L. (Alicante, Spain). This extruder has a screw diameter of 25 mm with a length/diameter (L/D) ratio of 24. The extruder has 4 heating zones, from the hopper to the die, where an ascending temperature ramp of 115–120–125–135 °C was applied. Once the different compositions were extruded, they were introduced into a pelletizer to obtain granules for subsequent processing by injection molding, using a Meteor 270/75 injector from Mateu & Solé (Barcelona, Spain), with the same temperature profile used in the extrusion process. The different compositions obtained together with their codification are summarized in Table 1.

### 2.3. Grafting Procedure

The grafting reaction was carried out in a mini mixer (HAAKE^TM^ PolyLab^TM^ QC, Thermo Fisher Scientific, Karlsruhe, Germany), together with the dicumyl peroxide (DCP) initiator. To obtain PBSA-g-IA, PBSA granules were started with PBSA granules that were physically mixed with IA and DCP at contents of 10 and 1 parts per hundred resin (phr) of PBSA, respectively. PBSA-g-MA was obtained with MA and DCP, following the same process and amounts as with PBSA-g-IA. The resulting mixtures were introduced into the mixing device and processed at 135 °C for 7 min. The resulting material was purified by refluxing in chloroform (Panreac S.A., Barcelona, Spain) for 4 h, and the hot solution was filtered and precipitated in cold methanol (Sigma-Aldrich S.A.). For the removal of unreacted or excess reagents, methanol was used to perform several successive washes, finishing the process with drying at 50 °C for 24 h in a CARBOLITE Eurotherm 2416 CG air circulation oven (Hope Valley, UK).

### 2.4. Characterization of PBSA/HEMP Composites

#### 2.4.1. Mechanical Characterization

The mechanical properties of neat BioPBSA pieces and PBSA/HEMP composites were evaluated for tensile properties, impact strength and hardness. Tensile tests were performed on an ELIB 50 universal testing machine from S.A.E. Ibertest (Madrid, Spain) on bone-type samples. The test was performed at a speed of 10 mm/min using a 5 kN load cell. The impact strength analysis was performed by Charpy impact test following ISO 179-1:2010 on 80 × 10 × 4 mm^3^ rectangular specimens notched on a Charpy pendulum from Metrotec S.A. (San Sebastian, Spain) with a 6 J pendulum. The hardness of the different compositions was obtained in a 76-D hardness tester from J. Bot Instruments (Barcelona, Spain), using the Shore D scale with a stabilization time of 15 s according to ISO 868:2003, taking measurements in different parts of a rectangular specimen of 80 × 10 × 4 mm^3^. All mechanical characterizations were performed at room temperature, and a minimum of 6 samples of each formulation were tested, and the values were averaged.

#### 2.4.2. Morphology Characterization

To study the morphology of the samples, the field emission scanning electron microscopy (FESEM) technique was used, using a ZEISS ULTRA 55 microscope from Oxford Instruments (Abingdon, UK). Samples used in the impact test were prepared, in which the breakage surface was treated by dispersing a thin layer of gold and palladium alloy in an EMITECH sputter coating SC7620 from Quorum Technologies, Ltd. (East Sussex, UK), to achieve a conductive surface, necessary for the correct operation of this technique. Subsequently, the samples were introduced into the microscope, with an accelerating voltage of 2 kV, with which 1000× images of the different samples were obtained.

#### 2.4.3. Infrared Spectroscopy

Chemical analysis was carried out by attenuated total reflection Fourier transform infrared (ATR-FTIR) spectroscopy. Spectra were acquired using a Bruker S.A. Vector 22 (Madrid, Spain) attached to a PIKE MIRacle™ single reflection diamond ATR accessory (Madison, WI, USA). The results data were obtained from the average of ten scans between 4000 and 600 cm^−1^ with a spectral resolution of 4 cm^−1^.

#### 2.4.4. Thermal Analysis

The study of the main thermal transitions of BioPBSA composites was determined by differential scanning calorimetry (DSC) in a Mettler-Toledo 821 calorimeter (Schwerzenbach, Switzerland). To guarantee the reliability of the tests, the average mass of the samples was kept at 5–8 mg, and they were placed in standard 40 µL aluminum crucibles. The samples were subjected to a thermal program that consisted of three steps, two heating cycles and one cooling cycle. The first heating cycle is destined to eliminate the thermal background trace due to processing and ranges from 20 to 130 °C, followed by the cooling cycle down to 0 °C, and finally a second heating cycle up to 250 °C. The heating and cooling rates were set at 10 °C/min. The tests were performed under an inert nitrogen atmosphere with a flow rate of 66 mL/min. The degree of crystallinity (*χ_c_*) was calculated using the following equation:(1)$$\%\chi_{c}=\left(\frac{{\Delta H}_{m}-{\Delta H}_{cc}\;}{{\Delta H}_{m}^{0}}\cdot \;\frac{100}{w}\;\right)$$
where ${\Delta H}_{m}{y\Delta H}_{cc}$ are the cold melting and crystallization enthalpies, respectively, and ${\Delta H}_{m}^{0}=$ 116.9 J/g corresponds to theoretical enthalpy of a 100% crystalline PBSA sample [41]. The term *w* represents the weight fraction of BioPBSA in the blend.

The thermal degradation and thermal stability of neat BioPBSA and composites were analyzed by thermogravimetric analysis (TGA) on a LINSEIS TGA 1000 thermobalance (Selb, Germany). The samples with an average mass of 15–20 mg were each placed in 70 µL alumina crucibles and subjected to a dynamic heating schedule from 30 to 600 °C at a heating rate of 10 °C/min in nitrogen atmosphere. All tests were carried out in triplicate to obtain reliable results.

#### 2.4.5. Thermomechanical Characterization

The thermomechanical properties of BioPBSA/HEMP were evaluated by dynamic-mechanical thermal analysis (DMTA) on a dynamic analyzer DMA1 from Mettler-Toledo (Schwerzenbach, Switzerland), working in single cantilever flexural conditions. Rectangular samples of dimensions 20 × 6 × 2.7 mm^3^ were subjected to a dynamic temperature sweep from 30 to 140 °C at a constant heating rate of 2 °C/min. The selected frequency was 1 Hz, and the maximum flexural deformation or cantilever deflection was set to 10 µm. A total of three tests per sample were averaged.

#### 2.4.6. Color and Wetting Characterization

For color measurement of samples, a Konica CM-3600d Colorflex-DIFF2 spectrophotometer from Hunter Associates Laboratory, Inc. (Reston, VA, USA) was used. Using this equipment, the coordinates (L*a*b*) were obtained according to the CIE L*a*b* (CIELAB) color space. L is the coordinate representing brightness L* = 0, darkness; L* = 100, lightness; a* represents the coordinate from green (a* < 0) to red (a* > 0); b* represents the coordinate from blue (b* < 0) to yellow (b* > 0). At least ten different measurements of the color coordinates were obtained and averaged. Contact angle measurements were performed with an EasyDrop Standard goniometer model FM140 (KRÜSS GmbH, Hamburg, Germany) that was equipped with a video capture kit and analysis software (Drop Shape Analysis SW21; DSA1). Distilled water was used as the test liquid. Wetting properties were evaluated on the surface of 80 × 10 × 4 mm^3^ rectangular samples. At least 10 water contact angle measurements were collected and averaged.

#### 2.4.7. Water Uptake Characterization

The evolution of the water absorption was studied using injection molded samples of 80 × 10 × 4 mm^3^, which were immersed in distilled water at 23 °C. The samples were removed and weighed weekly using an analytical balance with an accuracy of 0.1 mg, after removing the residual water with a dry cloth. The evolution of water absorption was followed for a period of 12 weeks. Measurements were performed in triplicate.

#### 2.4.8. Statistical Analysis

To measure the significant differences among the samples were evaluated at 95% confidence level (*p* ≤ 0.05) by one-way analysis of variance (ANOVA) following Tukey’s test. Software employed for this propose was the open-source R software (http://www.r-project.org), date of access: 5 May 2022.

## 3. Results

### 3.1. Mechanical Properties of BioPBSA/HEMP Composites

Mechanical characterization of BioPBSA composites are summarized in Table 2. The results indicate the effect on the mechanical properties of the different BioPBSA/HEMP composites. Neat BioPBSA showed a tensile modulus (E) of 281 MPa, a tensile strength of 21.1 MPa, and an elongation at break of 313.6%. These are typical values for this polymer, similar to those reported by other authors [42,43]. These values are indicative of a material with high ductility but with some stiffness. The incorporation of 30 wt.% of hemp fibers leads to a great increase in the tensile modulus (E), attaining a value of 3482 MPa and maintaining a tensile strength very similar to the neat BioPBSA, with a value of 19.6 Mpa, thus an improvement in the stiffness of the BioPBSA matrix can be observed. These values are remarkable, because the introduction of fillers, in general, reduces the tensile strength values of the composites in comparison with the neat polymer [44]. Roumeli et al. have observed the same behavior in high density polyethylene (HDPE) composites with hemp fibers [45]. With the addition of DBI to the base mixture, a reduction of the tensile modulus can be observed as the amount of DBI in the formulations increases, having tensile modulus values of 2405 and 1726 Mpa for the formulations with 7.5 *phr* and 15 *phr* DBI, respectively; with respect to tensile strength, the previous trend is repeated, with values of 16.3 and 12.7 MPa for the compounds formulated with a 7.5 and 15 *phr* DBI, respectively. This behavior could be due to the fact that the DBI produces some plasticization effect into polymer matrices [46]. The introduction of a plasticizer reduces tensile modulus and tensile strength, and at the same time the elongation increases with regard to the other compounds studied; all these phenomenon can be explained by several plasticization theories, such as the lubricity, gel and free volume theories [47,48]. With the introduction of PBSA-*g*-IA and PBSA-*g*-MA copolymers into the BioPBSA/HEMP, the tensile modulus and tensile strength results are 2505 MPa and 23.2 MPa, respectively, for the BioPBSA/HEMP/PBSA-*g*-IA composite, and 2748 MPa and 29.1 MPa for the BioPBSA/HEMP/PBSA-*g*-MA. These values may indicate an improvement in the interfacial adhesion between the polymer matrix and the natural fiber, as reported by El-Rafey et al. [49], as by preserving high tensile modulus values, an improvement in tensile strength is also achieved, surpassing even the polymer matrix.

On the other hand, the incorporation of the fibers randomly dispersed in the matrix causes a lack of homogeneity of the polymeric networks, due to the loss of cohesion between the polymer and the filler, which negatively affects the elongation at break values of the composites obtained [50]. This effect can be clearly seen in all PBSA/HEMP samples with an elongation at break (ε_b_) ranging from 5.2–8.2%, significantly lower than that obtained by the neat BioPBSA with an elongation of 313.6%. Other authors have reported similar behavior when incorporating natural fibers in polymeric PBS matrices [51,52].

Neat BioPBSA is a very ductile material, with a relatively high impact strength, 30.7 kJ/m^2^, obtained on notched test samples. Again, it can be seen how all the samples with hemp have reduced their impact strength values, yielding very similar values among the different samples, ranging between 9.5 and 10.6 kJ/m^2^, with this last result being obtained due to the incorporation of the PBSA-*g*-MA copolymer, as the incorporation of DBI achieves a certain plasticizing effect, although it does not produce any significant effect on the impact strength properties. This loss of impact strength is related to the presence of internal stresses due to the finely dispersed hemp particles incorporated in the matrix [53]. Several authors reported a similar decrease in impact absorbed energy when incorporating natural fibers in polypropylene and HDPE polymeric matrices [54,55].

Regarding the Shore D hardness, the results obtained indicated a great improvement in hardness, with respect to the neat BioPBSA and the different hemp composites. The hardness of BioPBSA stands at 56.2, while the addition of hemp fibers into the BioPBSA matrix promotes an improvement in the surface hardness of the composites, with values reaching 65.5 for BioPBSA/HEMP. The plasticizing effects provided by the DBI into the BioPBSA matrix are similar to those observed in the tensile modulus, with a decrease in hardness as the percentage of DBI in the composites increases, 60.4 and 54.6 for BioPBSA/HEMP/7.5DBI and BioPBSA/HEMP/15DBI composites, respectively. In relation to the composites compatibilized with PBSA-*g*-IA and PBSA-*g*-MA, their hardness values are close to those obtained in the PBSA/HEMP base composite.

### 3.2. Morphology of BioPBSA/HEMP Composites

The internal morphology of the composites is always related to their mechanical performance. Figure 1 gathers the field emission scanning electron microscopy (FESEM) images at 1000× magnification of the fractured surface of impact test samples from neat BioPBSA and BioPBSA/HEMP composites. Figure 1a shows the fracture surface of neat BioPBSA. The observed morphology is the typical rough surface ascribed to the plastic deformation of the polymer, which is indicative of its highly ductile behaviour [35], as it was stated in the mechanical properties due to an extremely high elongation at break. Figure 1b shows the addition of hemp fibers into BioPBSA. It can be seen that the adhesion and interaction between the lignocellulosic filler and the matrix is poor, due to the huge gap that exists between them. This is due to the filler being highly polar, while the matrix is mostly non-polar, promoting voids at the interface; this is indicative of some incompatibility between the polymer matrix and the fillers [56].The difference in polarity provokes a debonding phenomenon during the impact test, as was also observed by Burgada et al. [35]. This poor adhesion is responsible for the poor mechanical properties observed in the previous section. The addition of DBI (Figure 1c,d) reduced the gap between BioPBSA and hemp, which results in a better matrix–filler interaction. Additionally, the polymer surface seems to present higher roughness than the previous sample, which could be indicative of the plasticizing effect exerted by DBI, which corroborates the increase in elongation at break observed in the mechanical properties section. Finally, Figure 1e,f show the morphology of the blends with both compatibilizers, PBSA-*g*-IA and PBSA-*g*-MA, respectively. Their addition clearly improves the compatibility between the filler and the matrix. This fact is denoted by a practically inexistent gap between them. This is ascribed to the dual functionality of the compatibilizers. On the one hand, PBSA chains interact with the BioPBSA matrix. On the other hand, itaconic acid and maleic anhydride functionalizations interact with the hydroxyl groups present in hemp [57], thus leading to an improvement in the compatibility between the matrix and the loads, improving the interfacial adhesion between them, and providing a more homogeneous distribution of the applied external forces, which led to a higher mechanical strength [58].

All in all, these results perfectly match what was observed in the mechanical properties section, corroborating the plasticizing effect of DBI and the efficiency of PBSA-*g*-IA and PBSA-*g*-MA as compatibilizers, overcoming the difference in polarity between BioPBSA and hemp.

### 3.3. Chemical Properties of BioPBSA/HEMP Composites

Figure 2 gathers the FTIR spectra of all the developed BioPBSA/HEMP composites in the range 4000–600 cm^−1^. First, the FTIR spectra of neat BioPBSA shows the typical absorption peak at 1740 cm^−1^, corresponding to the stretching vibration of –OH and –C=O groups [59]. Other characteristic bands appear at 1451, 1080 and 1180 cm^−1^, which are related to C–O–C symmetric (1451 cm^−1^) and asymmetric (1080 and 1180 cm^−1^) stretching vibrations [60]. A very similar spectra for PBSA was observed by Seggiani et al. [61]. The mentioned peaks are present in all the blends, due to BioPBSA being the base material for all of them. When adding hemp to the blend, the main variation in the spectra is the appearance of a high intensity peak at 1200 cm^−1^, which is ascribed to the C–O stretch of the acetyl group in lignin present in hemp fiber [62]. This peak also appears in all the formulations with hemp, as it was expected. The incorporation of DBI does not provoke great modifications in the spectra. Nonetheless, a slight increase in the intensity of the peaks located in the range between 2800 and 3000 cm^−1^ is observed, attributed to the stretching vibration of the asymmetric stretching of the –CH and –CH_2_ groups [63,64]. The main PBSA peak at 1740 cm^−1^ seems to increase its intensity too, which could indicate higher concentration of C=O bonds present in DBI. Finally, the addition of PBSA-*g*-IA and PBSA-*g*-MA does not apparently vary the spectra of BioPBSA/HEMP. However, there is an increase of the intensity of the band at 1740 cm^−1^ in comparison with the non-compatibilized sample, which could be ascribed to the carbonyl groups of itaconic acid and maleic anhydride, indicating a certain degree of interaction between the compatibilizer and the blend [65].

### 3.4. Thermal Properties of BioPBSA/HEMP Composites

Figure 3 shows the results obtained in differential scanning calorimetry (DSC) tests for the second heating cycle of BioPBSA/HEMP composites. Additionally, the main thermal parameters are listed in Table 3. The first parameter to be analyzed is the melting temperature (T_m_) of the different samples. It can be observed that the introduction of hemp in the compounds does not produce significant differences in this parameter. The melting points of the composites ranged between 86.1 °C and 88.2 °C. Chiu et al. [42] obtained similar fusing temperatures in composites of PBSA with kenaf fibers, with T_m_ varying between 88–89 °C. The addition of a plasticizing agent such as DBI and compatibilizing agents such as PBSA-*g*-IA and PBSA-*g*-MA did not alter the T_m_ of the composites. On the other hand, the introduction of fibers did cause changes in the enthalpy of fusion ∆H_m_, due to the dilution effect [66], which affects the proportion of polymeric chains undergoing thermodynamic transition during melting.

Regarding to crystallinity, it can be observed how the degree of crystallinity of the neat BioPBSA and the composite with 30 wt.% of hemp fibers does not vary, with a value around 18.5% in both cases. According to the data reported by Dolçà et al. this is due to the amount of fiber introduced in the matrix polymer, increasing the crystallinity values as the fiber percentage increases [67]. With the addition of DBI to the BioPBSA/HEMP composite, the degree of crystallinity decreases slightly, leading to a combined effect of, to a lesser degree, plasticizing, predominating the compatibilization effect, increasing the polarity between the BioPBSA and hemp chains, resulting in a restriction in the mobility of the chains, with percentages of 16% for both formulations. With respect to the copolymers PBSA-*g*-IA and PBSA-*g*-MA, an increase in the percentage of crystallinity can be observed, exceeding the neat BioPBSA (18.5%) in both cases, with values of 22.1% for the BioPBSA/HEMP/PBSA-*g*-IA compound and 21.4% for BioPBSA/HEMP/PBSA-g-MA. This effect may be due to the better interaction between matrix and fiber provided by both compatibilizers, which favors the formation of crystallization nuclei, leading to a higher degree of crystallinity [68].

Figure 4 shows the mass percentage (Figure 4a) and the first derivative with respect to temperature (Figure 4b), corresponding to the study of the thermal stability and degradation of the BioPBSA/HEMP samples, while Table 4 gathers the main thermal parameter related to this test. Thermal degradation is highly dependent on the structure of the polymer and the additives and fillers it contains. Thermogravimetric analysis (TGA) allows the study of mass loss with increasing temperature. This mass loss is usually attributed to the scission of the polymer chain at high temperatures and to the volatilization of the low molecular weight components of the base polymer and of the additives used.

In this respect, the neat BioPBSA exhibited a one-step thermal degradation process, with T_5%_ and T_deg_ values of 335.9 and 383.1 °C, respectively. It can be observed that the incorporation of hemp into BioPBSA causes a decrease in thermal stability. These fibers are composed of 44.5% cellulose, 33% hemicellulose and 22% lignin [69]. These compounds have a degradation range that is occasionally lower than the degradation temperature of BioPBSA. Cellulose degrades thermally at 300–400 °C, hemicellulose at 220–315 °C, and lignin in the temperature range of 150–900 °C [70]. As a consequence, the thermal degradation of the composite starts at a lower temperature, with T_5%_ of 281.4 °C. With the addition of DBI to the composites, the T_5%_ is 220.9 °C for the BioPBSA/HEMP/7.5DBI composite and 191.9 °C for BioPBSA/HEMP/15DBI. This decrease in thermal stability is caused by the high sensitivity of DBI to temperature, evaporating as it reaches a temperature close to 200 °C, as happens with other compounds based on itaconic acid [71]. Thus, the composite with PBSA-*g*-IA has a similar thermal degradation behavior to the composite with 7.5 *phr* of DBI, with a T_5%_ value of 219.4 °C. On the other hand, the incorporation of PBSA-*g*-MA does not cause substantial changes in the T_5%_, being located in the same temperature range as the BioPBSA/HEMP base compound with a value of 278.9 °C.

With respect to the degradation temperature, T_deg_, a slight reduction of the values was observed in the composites, in a range of 360–366 °C, with a reduction of the degradation temperature varying between 17–23 °C, compared to the base polymer. This could be due to the thermal degradation of the hemp fiber components, which occurs over a wide range of temperatures, as mentioned above. The T_deg_ of the compatibilized composites show very similar results, indicating that the different additives added to the composites do not greatly affect the degradation temperature.

Regarding the residual mass, BioPBSA showed a mass percentage of 1.3%, while the hemp composites produced small residual amounts in the range of 2–3%. Similar values have been reported for blends with hemp fibers [69].

### 3.5. Dynamic-Mechanical Behaviour of BioPBSA Composites

Figure 5a,b show the curves of the evolution of the storage modulus (E′) and the damping factor (tan δ) as a function of temperature for BioPBSA samples and BioPBSA/HEMP composites with different compatibilizers. Table 5 summarizes the values of E’ at different temperatures, as well as the glass transition temperature (T_g_), obtained from the maximum peak in the dynamic damping factor diagram.

The thermomechanical behavior of BioPBSA was characterized by a E′ value of 2405 MPa at −110 °C. In the temperature range between 0 and 20 °C, the storage modulus decreases dramatically to 212.62 °C. This decrease in mechanical stiffness relates to the glass transition of the material, located at −37 °C and the low intrinsic stiffness of the base polymer [42]. By incorporating hemp fibers, an increase of the storage modulus is achieved in the different temperature ranges, as can be observed in the non-compatibilized BioPBSA/HEMP sample. This same trend is followed by the composites compatibilized with PBSA-*g*-IA and PBSA-*g*-MA copolymers, with slightly lower values for the BioPBSA/HEMP/PBSA-*g*-IA composite, and slightly higher for BioPBSA/HEMP/PBSA-*g*-MA. With respect to the T_g_ of the aforementioned composites, they are between −37 °C. In contrast, with the use of DBI as an additive, the T_g_ decreases to values of −42 °C and −45 °C as the proportion of DBI in both compounds increases. This decrease in T_g_ is an indicator of the plasticizing effect that DBI produces in the BioPBSA/HEMP base compound, as reported by Liu et al. [72]. The storage modulus for the BioPBSA/HEMP composites with DBI is characterized by the highest E′ results at −110 °C, while increasing the temperature to the 0–20 °C range results in E′ values below those of the other composites, with the lowest E’ values being obtained as the amount of DBI in the mixture increases. These results are in agreement with those previously observed in the mechanical properties.

### 3.6. Colour Measurement and Wetting Properties of PBSA/HEMP Composites

The results of the visual appearance of neat PBSA and PBSA/HEMP composites are shown in Figure 6. Generally, the introduction of natural fibers allows wood–plastic composites (WPC) to be obtained due to the intrinsic color of the lignocellulosic filler. In this case, the incorporation of short hemp fibers gives the composites a characteristic brown-like color [67].

Furthermore, the color coordinates of the CIELab space of each material have been measured, and these results are shown in Table 6. BioPBSA has a whitish color, with color coordinates L* = 76, with a*and b* showing values of −2 and 1.95, respectively. This is indicative of pure BioPBSA having low saturation shades between green and yellow. Similar color coordinates were reported by Liminana et al. [73] for PBS, with coordinates a* and b* very close to PBSA, but with a luminance L* of 85. With the addition of hemp fibers to virgin BioPBSA, a decrease in the luminance L* can be observed, with values around 44 in their composites. On the other hand, the a* and b* coordinates underwent a notable increase, especially the b* coordinate, with values between 17 and 22; on the other hand, the a* coordinate for the different composites increased with values from 7.7 to 9.9. These values in the a* and b* coordinates denote red and yellow shades, resulting in brownish colors, similar to natural woods, such as pine and oak woods [74]. As mentioned, all the obtained composites show similar values regarding the coordinates L*, a* and b*, maintaining the same percentage of hemp in each of the samples, so the addition of the different additives does not significantly affect the value of the color coordinates. This could be an advantage since the most suitable formulation could be used according to the chosen performance without affecting its characteristic color.

In addition to the visual appearance, the surface wetting properties have also been evaluated. Figure 7 summarizes the water contact angle (θ_w_) of BioPBSA and BioPBSA/HEMP composites. Low values of θ_w_ are representative of hydrophilic materials, while high values of θ_w_ imply low water affinity or hydrophobic behavior. In this aspect, a contact angle θ_w_ > 65° can be considered as the threshold of hydrophobicity [75]. Neat BioPBSA shows a θ_w_ of 73.2°, which is representative for the typical hydrophobic nature of polyesters, with a high contact angle value. According to the literature, the water contact angle of PBSA varies between 62° and 104° [63,76]. This variation may be due to the sample preparation process, which directly affects the surface roughness of the samples, a parameter that strongly influences the wettability of polymeric composites [76,77]. With the addition of hemp fibers, the different composites undergo a decrease in the contact angle, the most notable being that of the BioPBSA/HEMP base composite, dropping to values of 64°. On the other hand, all the BioPBSA/HEMP samples with additives show angles greater than 65°, showing a hydrophobic behaviour, which demonstrates the effectiveness of the additives with respect to the hydrophilic behavior of the BioPBSA/HEMP base composite.

### 3.7. Water Uptake Characterization

In general, the use of natural fibers in composites leads to an increased water absorption capacity, due to the high proportion of lignin, cellulose and hemicellulose they contain, since these compounds are highly hydrophilic [78]. This tendency to absorb water is one of the main drawbacks of wood–plastic composites (WPC), since excessive water absorption can reduce the mechanical properties of the final product, which is an impediment for certain industries and applications. Figure 8 shows the evolution of water uptake absorption of injection-molded composites for a total period of 12 weeks.

Neat BioPBSA has a very limited water uptake capacity, due to its hydrophobic behavior, with approximately 1 wt.% at 12 weeks of immersion in distilled water. Similar results were reported by Pei-lei et al. [79]. The BioPBSA/HEMP base mixture presents a maximum water absorption value of 7.5 wt.% in 12 weeks. Therefore, the addition of hemp fibers to the BioPBSA matrix significantly increases the water absorption capacity of the composites. This is due to the lignocellulosic nature of hemp fiber, which contains lignin, pectin, hemicellulose and cellulose, components with a high polarity that increase affinity for water [80]. The BioPBSA/HEMP/PBSA-*g*-IA and BioPBSA/HEMP/PBSA-*g*-MA composites yield water absorption results with values of 6.8 wt.% and 6.2 wt.%, respectively. This slight improvement is due to the functionalized groups of both formulations. On the one hand PBSA-*g*-MA presents some affinity for water due to the grafting of maleic anhydride, which has several highly hydrophilic oxygen-based groups [81], while itaconic acid (IA) presents polar carboxyl groups [82] with a less hydrophilic nature. In relation to the PBSA/HEMP formulations with DBI, the higher the DBI content, the lower the water absorption, with values of 6.2 wt.% for the formulation with a 7.5 *phr* of DBI and 5.2 wt.% for the formulation with a 15 *phr* of DBI. These results are related to improved fibre/matrix adhesion, significantly reducing water dispersion through the structure.

The results obtained here demonstrate the great water absorption capacity of BioPBSA/HEMP composites, and although this may seem a drawback in some applications, in other applications such as packaging, where the ability to absorb moisture from the food to preserve it is needed, this water absorption capacity becomes a great advantage [83].

## 4. Discussion

In this work, we have been demonstrated how the introduction of 30 wt.% of hemp fibers derived from agricultural waste has been used as reinforcement in BioPBSA matrices obtaining composites with a high environmental value. The obtention of the composites was assessed by means an extrusion process followed by an injection molding process to obtain the test samples. In terms of mechanical properties, the incorporation of fibers has allowed a large increase in the tensile modulus, starting from 2281–2748 MPa for the BioPBSA/HEMP base composite depending on the compatibilization strategy followed. The introduction of different additives helped to improve the different mechanical properties compared with the uncompatibilized composite. On the one hand, DBI promoted a plasticization phenomenon overlapped with an improvement of the compatibility between the polymer and the fiber, as observed in FESEM analysis. As a result, a reduction in the tensile modulus was observed but the best elongation at break results were obtained with respect to the base compound. On the other hand, the use of copolymers, such as PBSA-*g*-IA and PBSA-*g*-MA, managed to reduce the gap between the polymer and the filler, allowing improvement of the compatibility so that the tensile strength of the composites improved the neat polymer due to the reinforcement obtained with the natural fiber. With respect to thermal properties, the DBI-compatibilized composites show a decrease in terms of glass transition temperature and melting temperature with respect to the uncompatibilized compounds, while the other composites were not affected by the modifications proposed. In terms of thermal stability, the use of additive-based itaconic acid with a relatively low evaporation temperature led to a reduction of the thermal stability of the compatibilized composites; nevertheless, the initial degradation temperature is higher than the extrusion and injection molding working temperatures. With regard to the color study, when hemp is introduced, the composites acquire a brown color, which does not vary significantly with the addition of the different compatibilizers, acquiring a wood appearance, characteristic of WPCs. As could be expected, when adding a lignocellulosic filler, the contact angle decreases and the water absorption increases significantly with respect to the matrix polymer; although when adding the different compatibilizers, an increase in the contact angle can be observed, where the best results are obtained with the PBSA copolymers, and lower water absorption in DBI-containing composites. Overall, the results obtained with the different PBSA/HEMP compatibilized samples are really interesting, since the processing properties do not change too much, making it possible to use one material or another depending on the purpose of the application without affecting the visual aspect, an indispensable property in certain industrial sectors, such as rigid packaging. The employment of additives based on the itaconic acid has a great novelty since the polymer matrix employed and the filler are bio-based, so that the composites obtained are fully bio-based.

## 5. Conclusions

The use of new bio-based polymers, such as BioPBSA together with the addition of natural fibers from agroforestry industry residues, in this case short hemp fiber, allows the creation of novel environmentally friendly composite materials. The results of the mechanical tests indicate a great improvement in the tensile properties, and a decrease of the ductile properties; with the incorporation of the different additives this tendency is maintained, although with slight variations in the different mechanical properties, depending on the effect produced to the base compound (compatibilization or plasticization) that can be supported with the morphology results. The thermal properties obtained reveal negligible changes in the melting and glass transition temperatures, except for the compounds with DBI, which produced a decrease in these values. The DMTA results showed that the addition of hemp fibers and different compatibilizers resulted in an improvement of the storage modulus in the different temperature ranges studied. The use of compatibilizers also produced improvements in the water uptake properties, reducing the amount of water absorbed, as well as increasing the surface hydrophobicity of the compatibilized samples. Therefore, composites with attractive properties have been obtained, in addition to being environmentally friendly and helping to promote the circular economy.

## Figures and Tables

**Figure 1 polymers-14-01968-f001:**
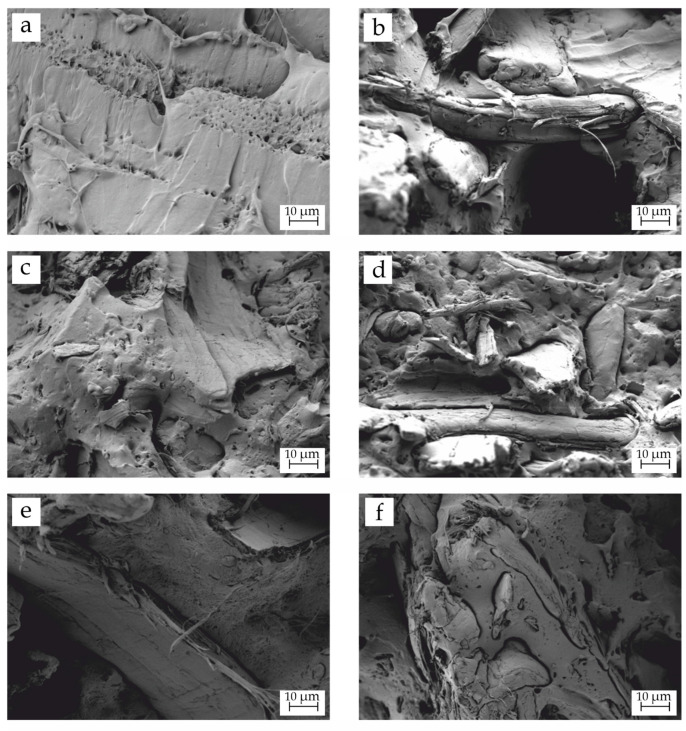
Field emission scanning electron microscopy (FESEM) images at 1000× of the fractured surfaces of (**a**) neat BioPBSA; (**b**) BioPBSA/HEMP; (**c**) BioPBSA/HEMP/7.5DBI; (**d**) BioPBSA/HEMP/15DBI; (**e**) BioPBSA/HEMP/PBSA-*g*-IA; (**f**) BioPBSA/HEMP/PBSA-*g*-MA.

**Figure 2 polymers-14-01968-f002:**
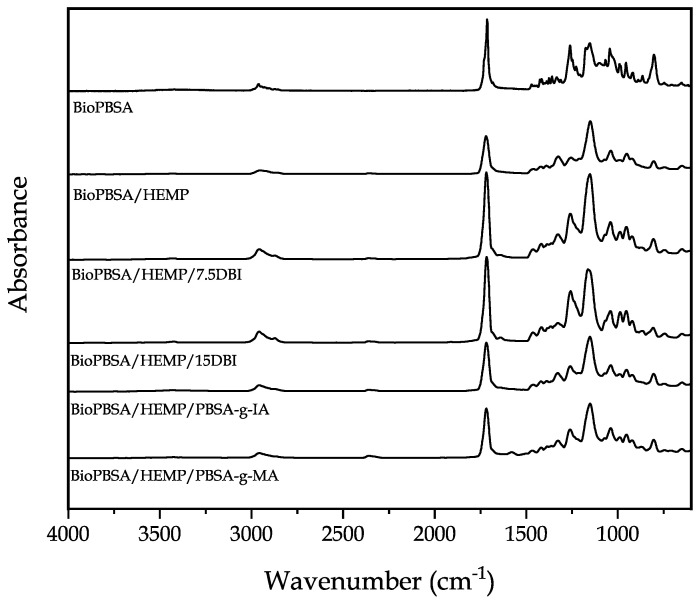
FTIR spectra of the uncompatibilized and compatibilized BioPBSA/HEMP composites.

**Figure 3 polymers-14-01968-f003:**
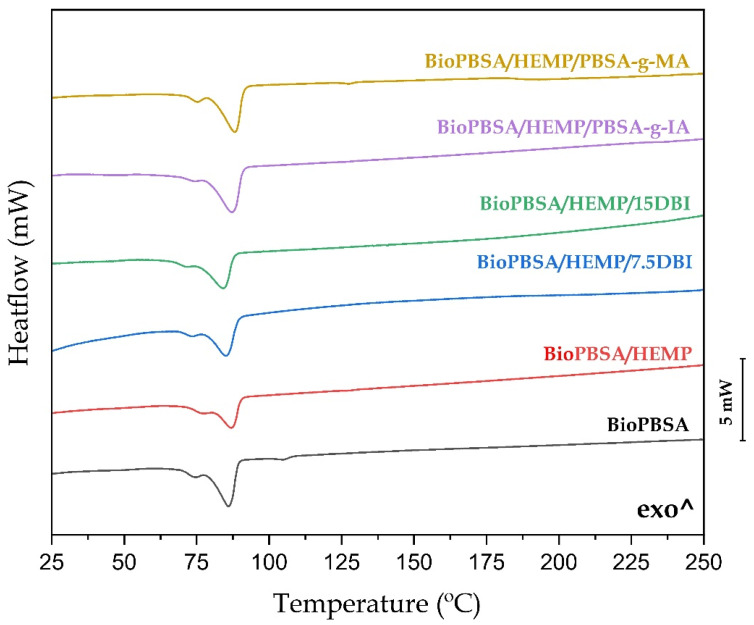
Differential scanning calorimetry (DSC) thermograms of BioPBSA/HEMP composites.

**Figure 4 polymers-14-01968-f004:**
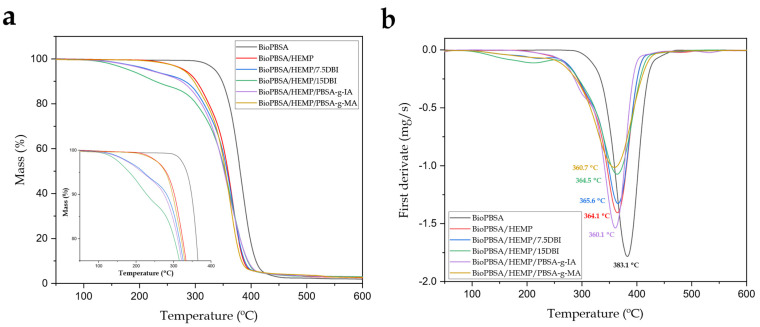
(**a**) Thermogravimetric analysis (TGA) curves and (**b**) first derivative (DTG) of BioPBSA/HEMP composites.

**Figure 5 polymers-14-01968-f005:**
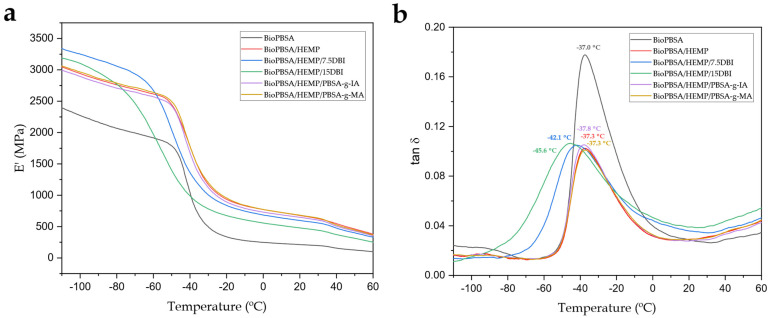
Plot evolution of (**a**) the storage modulus (E′) and (**b**) the dynamic damping factor (tan δ) of the injection-molded samples of BioPBSA/HEMP composites.

**Figure 6 polymers-14-01968-f006:**
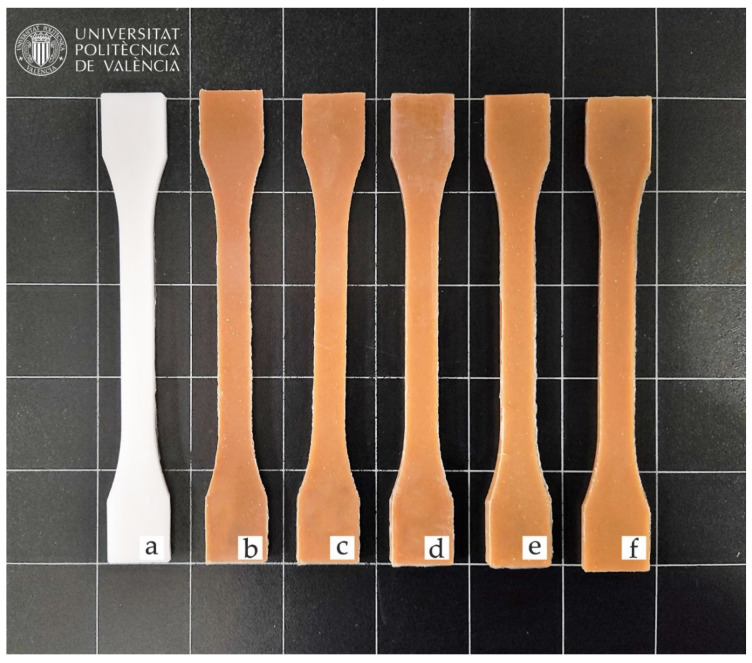
Visual appearance of the samples: (**a**) neat BioPBSA; (**b**) BioPBSA/HEMP; (**c**) BioPBSA/HEMP/7.5DBI; (**d**) BioPBSA/HEMP/15DBI; (**e**) BioPBSA/HEMP/PBSA-*g*-IA; (**f**) BioPBSA/HEMP/PBSA-*g*-MA.

**Figure 7 polymers-14-01968-f007:**
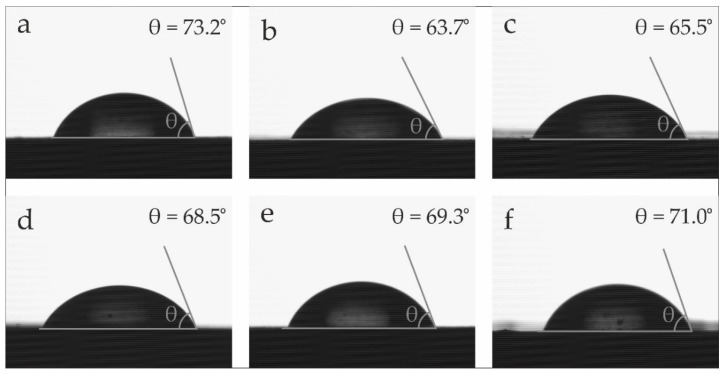
Water contact angle (θ_w_) of the samples: (**a**) neat BioPBSA; (**b**) BioPBSA/HEMP; (**c**) BioPBSA/HEMP/7.5DBI; (**d**) BioPBSA/HEMP/15DBI; (**e**) BioPBSA/HEMP/PBSA-*g*-IA; (**f**) BioPBSA/HEMP/PBSA-*g*-MA.

**Figure 8 polymers-14-01968-f008:**
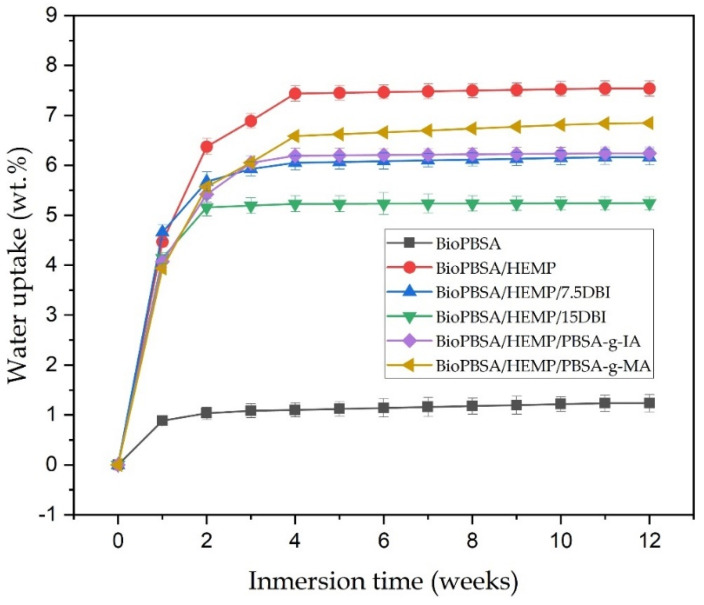
Water uptake of BioPBSA/HEMP composites after 12 weeks of immersion.

**Table 1 polymers-14-01968-t001:** Summary of compositions according to the weight content (wt.%) of BioPBSA/HEMP and parts per hundred resin (*phr*) of different compatibilizers.

Code	BioPBSA (wt. %)	HEMP (wt. %)	DBI (phr)	PBSA-*g*-IA (phr)	PBSA-*g*-MA (phr)
BioPBSA	100	0	0	0	0
BioPBSA/HEMP	70	30	0	0	0
BioPBSA/HEMP/7.5DBI	70	30	7.5	0	0
BioPBSA/HEMP/15DBI	70	30	15	0	0
BioPBSA/HEMP/PBSA-*g*-IA	70	30	0	5	0
BioPBSA/HEMP/PBSA-*g*-MA	70	30	0	0	5

**Table 2 polymers-14-01968-t002:** Summary of mechanical properties of the BioPBSA/HEMP composites in terms of tensile modulus (E), maximum tensile strength (σ_max_), elongation at break (ε_b_), Shore D hardness and impact strength.

Code	E (MPa)	σ_max_ (MPa)	ε_b_ (%)	Shore D Hardness	Impact Strength (kJ/m^2^)
BioPBSA	281 ± 3 ^a^	21.1 ± 1.0 ^a^	313.6 ± 6.1 ^a^	56.2 ± 0.8 ^a^	30.7 ± 1.2 ^a^
BioPBSA/HEMP	3482 ± 23 ^b^	19.6 ± 2.6 ^a^	5.2 ± 0.6 ^b^	65.5 ± 0.4 ^b^	9.5 ± 0.4 ^b^
BioPBSA/HEMP/7.5DBI	2405 ± 28 ^c^	16.3 ± 0.3 ^b^	8.2 ± 0.8 ^c^	60.4 ± 0.5 ^c^	9.9 ± 0.2 ^b^
BioPBSA/HEMP/15DBI	1726 ± 19 ^d^	12.7 ± 0.4 ^c^	8.2 ± 0.5 ^c^	54.6 ± 0.9 ^d^	9.5 ± 0.2 ^b^
BioPBSA/HEMP/PBSA-*g*-IA	2505± 12 ^e^	23.2 ± 0.9 ^d^	5.4 ± 0.3 ^d^	64.2 ± 0.5 ^e^	9.7 ± 0.3 ^b^
BioPBSA/HEMP/PBSA-*g*-MA	2748 ± 15 ^e^	29.1 ± 0.4 ^e^	5.8 ± 0.2 ^d^	64.7 ± 0.6 ^e^	10.6 ± 0.3 ^c^

^a–e^ Different letters in the same column indicate a significant difference among the samples (*p* < 0.05).

**Table 3 polymers-14-01968-t003:** Main thermal parameters of the composites with different amounts of hemp fiber in terms of melting temperature (T_m_), normalized melting enthalpy (∆H_m_), and degree of crystallinity (χ_c_).

Code	T_m_ (°C)	∆H_m_ (J/g)	χ_c_ (%)
BioPBSA	86.1 ± 1.2 ^a^	21.6 ± 0.9 ^a^	18.5 ± 0.2 ^a^
BioPBSA/HEMP	87.1 ± 2.4 ^a^	15.0 ± 1.1 ^b^	18.3 ± 0.5 ^a^
BioPBSA/HEMP/7.5DBI	85.2 ± 1.7 ^a^	13.3 ± 1.3 ^c^	16.3 ± 0.2 ^b^
BioPBSA/HEMP/15DBI	84.3 ± 1.3 ^a^	13.1 ± 1.4 ^c^	16.0 ± 0.3 ^b^
BioPBSA/HEMP/PBSA-*g*-IA	87.2 ± 1.5 ^a^	18.1 ± 1.1 ^d^	22.1 ± 0.3 ^c^
BioPBSA/HEMP/PBSA-*g*-MA	88.2 ± 1.1 ^a^	17.5 ± 1.4 ^d^	21.4 ± 0.4 ^c^

^a–d^ Different letters in the same column indicate a significant difference among the samples (*p* < 0.05).

**Table 4 polymers-14-01968-t004:** Main thermal degradation parameters of the BioPBSA/HEMP composites in terms of the onset degradation temperature at a mass loss of 5 wt.% (T_5%_), maximum degradation rate (peak) temperature (T_deg_), and residual mass at 600 °C.

Code	T_5%_ (°C)	T_deg_ (°C)	Residual Mass (%)
BioPBSA	335.9 ± 2.3 ^a^	383.1 ± 2.6 ^a^	1.3 ± 0.1 ^a^
BioPBSA/HEMP	281.4 ± 3.3 ^b^	364.1 ± 1.9 ^b^	2.5 ± 0.1 ^b^
BioPBSA/HEMP/7.5DBI	220.9 ± 1.7 ^c^	365.6 ± 1.6 ^b^	3.0 ± 0.2 ^c^
BioPBSA/HEMP/15DBI	191.9 ± 1.5 ^d^	364.5 ± 2.2 ^b^	3.0 ± 0.1 ^c^
BioPBSA/HEMP/PBSA-*g*-IA	219.4 ± 2.8 ^e^	360.1 ± 2.3 ^b^	2.7 ± 0.2 ^c^
BioPBSA/HEMP/PBSA-*g*-MA	278.9 ± 1.4 ^f^	360.7 ± 1.9 ^b^	2.7 ± 0.1 ^c^

^a–f^ Different letters in the same column indicate a significant difference among the samples (*p* < 0.05).

**Table 5 polymers-14-01968-t005:** Dynamic-mechanical properties of injection-molded samples of BioPBSA/HEMP composites, at different temperatures.

Code	E′ (MPa) at −110 °C	E′ (MPa) at 0 °C	E′ (MPa) at 20 °C	T_g_ (°C)
BioPBSA	2405 ± 25 ^a^	247.5 ± 0.7 ^a^	212.6 ± 1.5 ^a^	−37.0 ± 0.7 ^a^
BioPBSA/HEMP	3056 ± 28 ^b^	770.4 ± 1.0 ^b^	676.1 ± 0.7 ^b^	−37.3 ± 0.5 ^a^
BioPBSA/HEMP/7.5DBI	3336 ± 31 ^c^	682.0 ± 0.8 ^c^	595.8 ± 2.1 ^c^	−42.1 ± 0.7 ^b^
BioPBSA/HEMP/15DBI	3194 ± 30 ^c^	555.3 ± 0.6 ^d^	477.5 ± 2.7 ^d^	−45.6 ± 0.8 ^b^
BioPBSA/HEMP/PBSA-*g*-IA	3000 ± 21 ^c^	729.0 ± 0.7 ^e^	646.7 ± 3.1 ^e^	−37.8 ± 0.6 ^c^
BioPBSA/HEMP/PBSA-*g*-MA	3070 ± 23 ^c^	771.2 ± 0.5 ^e^	682.0 ± 3.9 ^e^	−37.3 ± 0.5 ^c^

^a–e^ Different letters in the same column indicate a significant difference among the samples (*p* < 0.05).

**Table 6 polymers-14-01968-t006:** CIELab color space measurements in terms of luminance and color coordinates (L*, a*, b*) of BioPBSA/HEMP composites.

Code	L*	a*	b*
BioPBSA	76.0 ± 0.1 ^a^	−2.0 ± 0.1 ^a^	2.0 ± 0.1 ^a^
BioPBSA/HEMP	43.6 ± 0.3 ^b^	7.7 ± 0.1 ^b^	20.9 ± 0.2 ^b^
BioPBSA/HEMP/7.5DBI	43.8 ± 0.1 ^b^	8.8 ± 0.1 ^c^	19.8 ± 0.1 ^b^
BioPBSA/HEMP/15DBI	44.0 ± 0.2 ^b^	8.7 ± 0.2 ^c^	19.1 ± 0.1 ^b^
BioPBSA/HEMP/PBSA-*g*-IA	44.1 ± 0.1 ^b^	7.7 ± 0.1 ^c^	17.7 ± 0.1 ^b^
BioPBSA/HEMP/PBSA-*g*-MA	44.3 ± 0.2 ^b^	9.9 ± 0.1 ^d^	22.1 ± 0.2 ^c^

^a–c^ Different letters in the same column indicate a significant difference among the samples (*p* < 0.05).

## Data Availability

Not applicable.

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
