# Peer review of "Green Composites from Partially Bio-Based Poly(butylene succinate-co-adipate)-PBSA and Short Hemp Fibers with Itaconic Acid-Derived Compatibilizers and Plasticizers"

_polymers, 2022, doi:10.3390/polym14101968_

Round 1

Reviewer 1 Report

The manuscript developed green composite polymers which is of interest by global researchers.

L21-25 and 25-27 are difficult to understand.

L25 “no important changes” is unclear whether what is considered as “important”

L61-62 Revise English

L76 Does hemp fiber have specific properties to be used with other bioplastics?

L98 Objective is unclear. It seems now this paragraph states what has been done in this work.

L121 What is coarseness value?

L28 How it was prehomogenized in a zip bag?

There should be statistical analysis in the Materials and methods and the results.

L243 Should be Results and discussion and the authors also discuss the results in this section.

L276 Add more discussion e.g., non-homogeneity of the polymeric networks decreased the elongation values of the polymers (https://doi.org/10.1002/app.45533)

L291 What is surface hardness? Did the samples form crust?

Table 2, 3 Add statistical analysis (and recheck the discussion about increase or decrease)

L313 Add more discussion e.g., Void space at the interface indicated incompatibility between polymer components (https://doi.org/10.1016/j.fpsl.2022.100844)

L319-320 Please reconsider this statement as the SEM cannot indicate plasticization effect.

L327 Improved compatibility between polymers in the blend matrices caused more homogeneous distribution of the applied external forces, leading to higher mechanical strength (https://doi.org/10.1016/j.foodchem.2021.131709)

L343 There are 3 wavenumbers. Which ones belongs to symmetric and asymmetric?

L402 Polymer components can be volatized at high temperature as well.

Page 11 Any reason for decreasing Tdeg. e.g., plasticization effects which increased molecular mobility and destabilized polymers? (https://doi.org/10.1016/j.lwt.2021.112356)

L513 Add more discussion e.g., Surface roughness also influenced by wettability of the polymer composites (https://doi.org/10.1016/j.meatsci.2020.108367)

L543 Void space also enhanced diffusion of water through matrices.

L554 Should this be conclusion? If so, it should be shortened and state key findings only. Avoid repeating the results.

Reviewer 2 Report

In their research article titled "Green composites from partially bio-based poly (butylene suc- cinate-co-adipate)-PBSA and short hemp fibers with itaconic
acid-derived compatibilizers and plasticizers" the author have been investigated the bio-based polymeric matrix such as BioPBSA and the introduction of a 30 wt.% short hemp fibers as a natural reinforcement to obtain materials with maximum environmental efficiency. Different additives derived from bio-based itaconic acid have been added to the BioPBSA/HEMP composite, such as Dibutyl
itaconate (DBI) and a copolymer of PBSA grafted with itaconic acid (PBSA-g-IA). In addition, copolymer of PBSA grafted with maleic anhydride (PBSA-g-MA) was also tested. The authors confirmed the effectiveness of these strategies by chemical, mechanical, morphological, thermally, thermomechanical, and FTIR characterization. However, the motivation of the study is sufficient enough. I appreciate their efforts; however, modify your manuscript according to the following changes. I think it could be published with major revisions.

Q1- the author needs to improve the material, method, and characterization section; most of the text is similar to previous studies. Or may give only references also fine. 

Q2- in the introduction part, paragraph 1, most of the sentences or statements reference are missing and do not cites accordingly; please carefully check. 

Q3- in the introduction and results discussions, the author often uses the word  (Other authors have). Please avoid repetition and use the author's name.

Q4- With the addition of DBI to the base mixture, a reduction of the tensile modulus can be With the addition of DBI to the base mixture, a reduction of the tensile modulus can be ues of 2405 and 1726 MPa for the formulations with 7.5 phr and 15 phr DBI respectively, concerning tensile strength the previous trend is repeated, with values of 16.3 and 12.7 MPa for the compounds formulated with a 7.5 and 15 phr DBI, respectively. This behavior could be since the DBI produces some plasticization effect in the polymer. Please make a quantitative compression with previously reported polymers. 

Q-5 The conclusion or discussion is very long; please make it short and add some quantitive or qualitative data, making it easier for the reader.

Q-6 plagiarism of the manuscript is also higher than polymers standers.

Reviewer 3 Report

Dear authors,

Article structure is prepared according to journal Polymers and author instructions.  

The topic is in the scope of the journal Polymers, it clearly presents green composites that have been developed and characterized using a bio-based polymeric matrix and hemp fibres.

Here are my comments:

1) Abstract: clearly presents the manuscript review.

2) Introduction provides sufficient background and includes all relevant references. The references are cited correctly. The aim and the goal of the research are clearly presented.

3) Line 360: Figure 2: peaks should be added to certain spectra. Namely, authors are explaining the peaks around 2900 cm-1 and at certain samples, there are more than 1 peak at the area.

4) The conclusion: is well presented.

5) References: are correctly cited and written.

Overall, the manuscript is very well presented and described.

Round 2

Reviewer 1 Report

The manuscript has been revised and improved.

Reviewer 2 Report

The authors perform good quality work for the scientific community, and the authors explain the results with a mechanism clearly. I appreciate their efforts; I think it could be published.